# Domains and Categories of Needs in Long-Term Follow-Up of Adult Cancer Survivors: A Scoping Review of Systematic Reviews

**DOI:** 10.3390/healthcare12111058

**Published:** 2024-05-22

**Authors:** Nicolas Sperisen, Dimitri Kohler, Nicole Steck, Pierre-Yves Dietrich, Elisabetta Rapiti

**Affiliations:** 1Institute of Global Health, Faculty of Medicine, University of Geneva, 1205 Geneva, Switzerland; 2Swiss Cancer League, 3001 Bern, Switzerland; dimitri.kohler@gmail.com (D.K.); nicole.steck@krebsliga.ch (N.S.); 3Clinique des Grangettes, Hirslanden, 1224 Geneva, Switzerland; pierre-yves.dietrich@hirslanden.ch; 4Faculty of Medicine, University of Geneva, 1205 Geneva, Switzerland; 5Geneva Cancer Registry, Institute of Global Health, Faculty of Medicine, University of Geneva, 1205 Geneva, Switzerland; elisabetta.rapiti@unige.ch

**Keywords:** cancer survivors, needs, follow-up, supportive care, information, health system, assessment

## Abstract

The number of long-term cancer survivors increases continually. Understanding their needs is crucial to ensure an adequate follow-up. The aim of our study was to summarize the current literature concerning needs and what influences these needs. A scoping review of systematic reviews was conducted according to the recommendations of the Joanna Briggs Institute. Four electronic databases were searched. Of 414 retrieved papers, 11 met the eligibility criteria. Needs were aggregated into six domains (health-related information, health system, mental, practical, relationship and physical) and 15 categories. The lack of adequate information and the lack of access and/or continuity of supportive care were the most prominent needs. Female gender, younger age, a low level of family and/or social support, and higher educational level were identified as risk factors. Employment and relationship status can affect the needs both in a positive and negative way. The weeks or months after the end of the treatments are particularly critical, and needs can be emphasized during this period. The experience of cancer could also lead to positive changes. The variety of needs affects the quality of life of cancer survivors. Needs assessments should be systematically provided to ensure a better awareness of health professionals and to allow an individual, holistic, and integrated follow-up.

## 1. Introduction

### 1.1. Rationale

In 2022, about 20 million people worldwide received a new cancer diagnosis [1]. With the growth of the population, increased life expectancy and the persistence of risk factors (exposures), the trend is estimated to increase up to 32.6 million in 2045 [1]. Cancer is the second cause of death worldwide, just behind cardiovascular diseases [2]. However, thanks to the progress in cancer screening and medical treatments, the mortality rate regularly decreases, and more patients can benefit from a better long-term survival. As a result, an increasing number of people are living with or after cancer. By the end of 2022, approximately 53.5 million people worldwide were estimated to be cancer survivors [3].

Mullan, with his “Seasons of survival” [4], and Miller [5], with a subsequent adaptation, have played a major role defining the several phases of the cancer journey and the range of concerns unique to each of these one. Among these phases, the transition to survivorship is not trivial and particularly important. Cancer survivors must reintegrate into daily life and establish a new normality, while they lose the “daily structure and the routine of cancer treatments” [6]. Consequently, the end of acute care represents a “Turning point”, accompanied by some important changes and multidimensional distress, but could also lead to “a profound personal growth and transformation” [7]. For Derbez and Rollin, cancer survivors should manage the consequences of the illness in “all spheres of their existence” [8]. In the long term, the social and/or professional activities, as well as the quality of life, could be affected. A meta-analysis [9] found a significant impact on quality of life up to 26 years after cancer diagnosis. It is crucial to acknowledge that each survivor’s journey is unique and may vary in timing. Factors such as the type of cancer, the individual’s bio-psycho-social characteristics, and their life circumstances play significant roles [10]. Therefore, health professionals should recognize this complexity to provide holistic and tailored support.

### 1.2. Objectives

The needs of cancer survivors can be diverse and vary in duration, therefore supportive care should be personal and proposed at bio-psycho-social levels. The American Institute of Medicine and National Research Council published 10 recommendations to optimize the provision of supportive care. Two of them emphasize the need to base care on peoples’ needs, values and preferences, and the need to anticipate these factors [11]. To identify the needs of cancer survivors, we conducted a scoping review by addressing the following research questions:What are the needs in long-term follow-up of cancer survivors?Are there socio-demographic differences in the needs (e.g., age, gender, marital status, income, location, etc.)?Are the needs greater in the transition phase (directly after acute treatments)?

## 2. Materials and Methods

This scoping review was conducted according to the principles of the approach (recommendations) developed by the Joanna Briggs Institute (JBI) [12] for scoping studies. A protocol was written but not published. It is available on request. The PRISMA extension for scoping reviews checklist (PRISMA-ScR) was used for reporting items [13].

### 2.1. Eligibility Criteria

This study used the population, context, concept (PCC) framework recommended by the JBI [12]. The “population” was defined as adult cancer survivors. We decided to exclude people who had received a pediatric or adolescent (<18 years) cancer diagnosis. Due to different stages of physical, psychological, and social development, this section of the population has its own needs. For example, the needs for sexual health can be very different from those of adults [14]. There is no consensus on the definition of “Cancer Survivor”, and self-identification with this group is very personal and depends on many factors [15]. We used the definition of the European Organisation for Research and Treatment of Cancer (EORTC): “a cancer survivor is an individual who was diagnosed with cancer, finished primary cancer treatment and has no evidence of active disease” [16]. The reason was that we wanted to focus on the “post-cancer” period and deal with the chronic aspect of the disease. To ensure we have the right population, we excluded survivors where the time since diagnosis was less than two years. The “concept” was to consider the follow-up needs, domains, and categories (e.g., information, work issues and emotional factors), and not the specific supportive care needs (e.g., physiotherapy or psycho-oncology). Finally, the “context” was high-income countries. We excluded studies focusing on the indigenous populations of countries, as this was not relevant for Switzerland.

Inclusion criteria were:Study design: Systematic reviews;Published in peer-review journals;Languages: English, French and German;Focusing on cancer survivors:○All kinds of cancer;○Male or female;○Curative intent;Studies from high-income countries.

Exclusion criteria were:Other study designs, conference proceedings (in the absence of a full-text paper);Publications older than 2011;Focusing on cancer survivors <18 years old (at diagnosis);Time since diagnosis <2 years;Studies focusing on specific supportive care needs (e.g., physiotherapy);Studies from low- and middle-income countries (LMIC);Focusing on the indigenous population of high-income countries (HIC).

### 2.2. Search Strategies

Four electronic databases were searched with the help of a professional librarian information specialist: PubMed, Embase, Cochrane Library and Epistemonikos. The search was concluded on 3 January 2023.

The search was based on three key concepts: unmet needs, follow-up, and cancer survivors. To increase the sensitivity of the search, controlled terms were associated with free terms (see Appendix B). The search strategy for PubMed is described in Appendix C. The other search strategies are available on request.

### 2.3. Selection of Sources of Evidence

The elimination of duplicate references was performed in the reference management software MENDELEY 2.107.0 (New York, NY, USA). The selection of studies that meet the inclusion/exclusion criteria was performed using RAYYAN in three distinct steps:(a)Selection based on titles: the first author (N.Sp.) selected based on the titles of the articles. A second author (D.K.) reviewed the titles of excluded articles, and any differences of opinion were discussed between them;(b)Selection based on abstracts by two independent reviewers (N.Sp., D.K.). The results were compared and any differences in the selection of the reviewers were discussed on a case-by-case basis. Reasons for excluding studies were reported;(c)Selection based on the full texts by the first author (N.Sp.). If there were any doubts, the lead author discussed them with the other authors of the review. Reasons for excluding studies were reported. The reference lists of the selected studies were searched for further studies.

### 2.4. Data Charting Process

The first author (N.Sp.) extracted data from the selected studies using a form with specified categories and sub-categories. A co-author (D.K.) controlled the extraction of the data, and completed any missing items if necessary.

### 2.5. Data Items

The following variables were reported:Source: title, author(s), year, journal, DOI/ISBN;Characteristics of the study: number of included articles, objectives, database consulted, inclusion/exclusion criteria, bias, results of the quality appraisal;Population: age, gender, cancer type, number of participants;Concept: domains of needs;Context: country, period (of the studies), setting;Results: needs, unmet needs, conclusion, recommendation for screening tool, influence of the comorbidities, influence of socio-demographical factors, information about the transitional phase.

### 2.6. Critical Appraisal of Individual Sources of Evidence

Analog to Ava Lorenc and his colleagues [17], the quality of the individual systematic reviews was assessed with the Assessing the Methodological Quality of Systematic Reviews (AMSTAR) checklist [18]. We judged the validation of 13 of the 16 criteria. The three last criteria are only used for meta-analysis. We awarded two points for each criterion that was met and one for those partially met. Studies are considered to be of sufficient quality if they achieve a score of 10 out of 26. Below this level, they are not included in the results. Above 15 points, studies are considered to be of good quality.

### 2.7. Synthesis of the Results

Due to the heterogeneity of the studies, a narrative analysis was used to identify the domains and categories of needs for each study. If any of the studies included in the selected systematic reviews did not meet the PCC criteria, they were excluded from the synthesis of results.

A map with the identified needs was created. Needs were later aggregated by domains and categories. But, due to the heterogeneity of the studies (population, measuring instruments used), it was not possible to find a consensual basis. For this reason, six new global domains and 15 sub-categories of needs were created. These domains were ranked by counting the number of studies reporting the individual needs.

## 3. Results

### 3.1. Selection of Sources of Evidence

The search returned 414 results (Figure 1). After de-duplication, 373 were screened by title and abstract. In total, 356 of them were removed, mainly because of the wrong outcome. A total of 17 articles’ full text were read, and 10 were included. In the reference lists of these, one new paper fulfilling all criteria was found. In total, eleven articles were included in this scoping review.

### 3.2. Characteristics of Sources of Evidence

All reviews were published between 2013 and 2021 (Table 1). These reviews included publications ranging from 1990 to 2021. The main countries analyzed in the reviews are USA (*n* = 8), Canada (*n* = 7), UK (*n* = 6), Australia (*n* = 6) and the Netherlands (*n* = 4). The rest of the countries are spread all over the world, but are mostly in Europe and Asia. Some studies focus only on one cancer type: colorectal cancer (*n* = 3), gynecological cancer (*n* = 2) and thyroid cancer (*n* = 1). The others (*n* = 5) examined three or more cancer types. Five studies mentioned their source of funding [14,19,20,21,22], and none of them disclaimed conflicts of interest.

### 3.3. Critical Appraisal within Sources of Evidence 

According to our quality appraisal (Table 1), four of the included studies [19,20,24,27] were considered as good quality, with a score of 15 or more, and six studies [14,21,22,25,26,28] were considered as sufficient quality, with a score between 10 and 14. The quality of the last one [23] was not sufficient (6 out of 26), and was not included in the results.

### 3.4. Results of Individual Sources of Evidence 

Some of the included studies did not fully meet the PCC criteria. But for all of them, the results are found in other studies meeting the PCC criteria, or it is possible to differentiate the studies within a systematic review and, thus, exclude the results related to non-PCC studies. This distinction is not possible in the study of Lehmann et al. [14]. But, as this study is the only one to explore sexual health-related needs of cancer survivors, we decided to keep it. The results of each included study for the three research questions are presented in Appendix A Results According to Research Questions.

### 3.5. Synthesis of Results

Table 2 provides an overview of the needs. As previously mentioned, they were aggregated and compiled into six domains: health-related information, health system, mental, practical, relationship and physical. For enhanced precision, the needs were subsequently classified into 15 distinct categories. Examples are provided within each category for clarity.

#### 3.5.1. Health-Related Information

This need was reported in all the included studies. To meet certain objectives, cancer survivors should receive and process adequate information on all types of subjects. Timely [25,27], repeatedly throughout-follow-up [25], and tailored information [19,20], especially about the short- and long-term effects of cancer and/or treatment was particularly needed [14,19,20,21,24,28]. Information on aftercare, support or rehabilitation services was also requested [19,20,21,24,26,28]. Nevertheless, it should be emphasized that the need for information is not only for the survivorship phase, but is also perceptible from the very first screening tests and diagnosis, particularly in relation to the disease and the treatment [19,20,21,24,27,28]. Cancer survivors needed help to process the information when their cognitive skills were not sufficient. Pape et al. [27] recommended first assessing the person’s level of understanding. Moreover, it was recommended to provide written information in addition to oral information delivered in the consultations [25,27]. To optimize the delivery of information, Maguire et al. advised to systematically assess the impact of the information received [20]. For their part, Kotronoulas and al. encouraged health professionals for better patient education [19].

#### 3.5.2. Health System

As far as the health system is concerned, one of the most prevalent reported needs was related to the access and/or continuity of care and supportive care [20,22,25,26,27,28], which can intensely influence the experience of people with cancer, particularly at the time of diagnosis [20]. The need for better coordination and communication among healthcare professionals, especially between primary and secondary care, was underlined [19,28]. Care navigation [28] and overall post-treatment follow-up (mainly by specialist nurses) was also lacking [19]. Health and social professionals have a role to play in optimizing care for cancer survivors. They should treat the person as an individual and not as a case [19]. They should be good listeners, trustworthy, and sensitive (empathetic) to the emotions of patients and those around them [14,19,20]. Cancer survivors wished that GPs make pro-active contact and wanted to discuss with them how to manage and adjust to life after treatment [24].

#### 3.5.3. Mental

Cancer can cause some mental disruptions in the fields of existential, emotional, interpersonal/intimacy, and psychic health that can impact a person’s behavior or reasoning. The most prominent concern in this domain was the fear of cancer recurrence and progression [19,20,22,25,26,28]. This persistent need [20,25] has a negative impact on quality of life and emotional well-being [20]. To avoid or limit the stress of cancer survivors, the feeling of being abandoned [19,27,28] should be addressed. This is also true for the “need for reassurance about being treated, especially when the safety net of the treatment ends” [22]. Managing anxiety was a need that was regularly highlighted in the psychic field. It was strongly correlated with a high level of unmet needs, irrespective of the domain [22,26].

#### 3.5.4. Practical

Cancer can affect the daily life of survivors and their relatives. To improve this situation, support was needed for transportation [19] or travel [28], for commitment to maintaining or adopting healthy behavior [19,25], and for daily activities [22,25,26] like mowing the lawn, washing the car, cleaning the house, or cooking. Out-of-pocket costs for cancer treatment, costs for care and symptom management, as well as indirect costs (travel, loss of earning, etc.) could be a financial burden for cancer survivors [19,20,22,24,25,26,28]. For this reason, keeping a job or returning to work was a key issue. It could be deteriorated among other by the fatigue or some barriers at the office [25]. Cancer survivors needed help with this [19,20,25].

#### 3.5.5. Relationship

The illness and/or treatments can disrupt interactions with the family and the social environment. The shame or embarrassment felt in certain social situations [20,25] could lead to isolation [20,28]. Support from the family, friends or peers ensured that survivors do not feel abandoned [25,27,28]. Access to support groups was another good strategy [19,28]. Survivors were also concerned about the well-being of their family and loved ones and the impact of their illness on them [20,26,28]. In this way, support for the family with regard to their own worries for the survivors [19] could be necessary.

#### 3.5.6. Physical

Disease and treatments can lead to bodily and cognitive consequences. The most notable need was help to cope with lack of energy or cancer-related fatigue [19,20,22,25,26,27]. Other frequently cited needs were dealing with pain [19,20,22,25], urinary inconsistence [20] and bowel dysfunction [20,25]. Cancer survivors also needed help to manage side effects that can affect their sexual lives [14]. Cognitive dysfunctions like memory loss or difficulties to concentrate are less visible, but still require support [20]. Although physical problems were not the most frequently mentioned in the selected studies, they were particularly important and needed to be managed because they could influence other areas [25]. For example, they were the cause of major functional limitations that could impact the smooth running of daily life. Changes in physical capacity could also affect the psychological/emotional domain, leading to increased stress, psychological problems, frustration, and uncertainty about the future. Moreover, these disorders were frequently associated with the recurrence or progression of cancer [25].

#### 3.5.7. Socio-Demographic Factors

Five [14,19,22,26,28] of the studies selected investigated the influence of socio-demographic factors on needs. Female gender [14,19], younger age [14,19,22,26,28], less family or social support [19,22,26] and higher education level [19,22,26,28] were considered risk factors. Cancer survivors with a higher level of education were also found to be more depressed or have an abnormal fatigue score [28]. Needs were influenced in a different way according to employment status and relationship status. Unemployed people had needs mostly in the domains of health care and information for financial support [22], while employed people needed emotional support, for example [19]. For sexual health-related care needs, single people expressed different needs compared to those with partners [14]. Single persons have more needs for support in dating new partners, while persons who are in partnership need support to discuss frankly about sex [14]. For geographical location, Van der Kruk et al. [28] found that people living in rural areas had certain specific needs. These were related to access to services or the lack of available services. The more isolated their place of residence, or the further away they were from major urban centers, the more these needs were exacerbated [22,28]. Moreover, people living in rural areas were more likely to experience problems related to finance, transport, and separation with family than people living near or in urban centers. They also reported worse social and emotional outcomes than urban survivors [28]. It should be noted that most of the studies included in this systematic review were conducted in large countries (USA, Canada, and Australia), where, due to the distances, the rurality context is different from other countries.

#### 3.5.8. Transition Phase

The weeks or months following the end of acute treatment are particularly sensitive for cancer survivors. As they receive less support than during the treatment period and begin to feel the first side effects of treatment or disease, they must proceed with their lives. In addition to frequent feelings of abandonment [19,25,26], many survivors were anxious to leave the hospital system [26], or should be reassured to continue to be treated [22]. Seven studies [14,19,20,22,24,25,26] found greater needs during this period. Coordination among healthcare professionals [19], a proactive approach by general practitioners [24], communication with the treatment team [20], help with specific symptoms [26], long-term support [25], or transmission of information on late effects and supportive care [19,22] were the main needs. The fear of cancer recurrence was not influenced by the period, and was just as important throughout the patient’s journey [20,25].

## 4. Discussion

### 4.1. Summary of Evidence

Cancer survivors are a very heterogeneous group, with a large number and variety of support needs. Our review identifies the main needs experienced by cancer survivors. As these needs are described differently from one study to another, we decided to aggregate them into 15 categories, divided into six dimensions. Apart from the health system domain, which is more on a meta-level, our results match those identified by Margaret Fitch almost 20 years ago [29]. In her paper, she said: “Cancer and its treatment have an impact on individuals that is felt in a number of ways”. She proposed a supportive care framework for cancer care, presenting examples of needs for cancer survivors and categorizing them in seven domains: physical, psychological, spiritual, emotional, practical, social, and informational. This review shows that although these needs have been known about for a long time and have changed little over time, they remain problematic, and are not yet adequately addressed.

One of the most important needs identified in our review is associated with processing health-related information.

According to Sørensen et al. [30], processing information requires four types of skills: (i) being able to access the information; (ii) understanding; (iii) assessing its quality in terms of reliability, level of evidence, and potential commercial conflicts; and (iv) being able to apply the information to health-related decisions. This allows the person to take an active role in the management of the illness, leading to a better quality of life. Two of these points are highlighted in our review. First, survivors must have access to good quality and adequate information. When information is not provided, survivors feel “disempowered and unimportant” [31]. More specifically, this information must be clear, accurate and reliable. It must also be provided in good time, in appropriate quantities, and in a sensitive manner. This last point depends mainly on the healthcare professional’s communication behavior [14]. Secondly, survivors should have certain cognitive abilities to understand this information. In Switzerland, around one out of five (18.48%) cancer survivors sometimes, and one out of twenty (5.5%) often or always, has difficulties understanding health-related written information [32]. Health professionals should therefore assess the understanding and the impact of the received information and provide support if necessary.

Cancer can lead to a variety of specific personal needs. These overlap and are interconnected and, for this reason, need to be considered as a whole. Mental health needs are one of the most important. The first six to twelve months after treatment is a critical period for depression and anxiety [33]. In addition, during this period, the quality of life could be affected by the lack of “accessible professional counselling within the hospital framework” [33]. Survivors also need help to deal with uncertainty about the future and difficulty managing adversity, lifestyle changes or the fear of dying. In the emotional sphere, needs mainly concern self-image and the difficulty of managing the associated changes. At the interpersonal/intimacy level, survivors must deal with changes in sexuality, whether due to dysfunction or loss of desire. Relationships with partners can also be difficult if their desires prove to be problematic for the person affected by mental disorders. Many other reasons can affect relationships and lead to avoiding social contact. For example, embarrassment due to the presence of a stoma and the associated smell [27], difficulties in dealing with tensions, or difficulties in managing the social role change. Interactions with family or communication with children could also be a source of preoccupation for the cancer survivor, as well as the inability to ask for help or find support for the family’s own worries. Accessing support groups is one of the identified solutions in our review. According to the Macmillan website [34], one of the benefits of the peer support groups is to share experiences and find coping strategies with people who have a similar background. This could be performed face-to-face, online or by telephone. However, it is important to understand that such groups are not for everyone due to some barriers, such as embarrassment about sharing experiences, or the sensitivity of some peers to death [35]. Daily life is also impacted by cancer. Cancer survivors have needs regarding transportation and access to care, and face challenges with daily tasks such as housework, childcare, and gardening. They also have a greater risk of facing financial difficulties than the general population [36]. Cancer survivors need support for dealing with the financial burden and distress (financial toxicity) [36,37] of the disease and/or the treatment, to improve their financial well-being. A way of limiting the financial burden is to ensure that the person can keep an income. In France, the VICAN5 study shows that 20% of people aged between 18 and 54 who were working at the time of diagnosis are no longer working five years later. This loss of employment primarily affects the most vulnerable people in the labor market. Most people stopped working 3 to 5 years after diagnosis, indicating a medium-term effect of the disease [38]. A progressive return to work is recommended, and some adaptations (e.g., position, working hours, workload, etc.) are often required. Sometimes a change of activity is the only solution. Among people employed at time of diagnosis, 54.5% have kept the same job and 17.4% have changed [39]. The role of the employer is important in optimizing the work environment [40], particularly in managing the reactions of colleagues and encouraging the integration of the person concerned. Daily health behaviors are also impacted. In total, 53% of French cancer survivors declare having reduced or stopped physical activities, compared to 34.3% who do not change anything, and 12.7% who train more [41]. Less than one person out of two (39.8%) stopped smoking within five years after diagnosis [42]. Physical inactivity and smoking are negatively associated with the health-related quality-of-life of cancer survivors [43]. Health promotion must be integrated into the follow-up of cancer survivors [9,44,45,46], and special attention is needed to provide them with support to maintain or integrate healthy behaviors in their daily life. Finally, cancer survivors may experience cognitive problems such as memory loss, concentration difficulties or attention problems. They may also experience physical problems that affect all parts of the body, and are often linked to the type of cancer. Cancer-related fatigue (CRF) or a lack of energy are common symptoms among survivors, which affect the daily activities and well-being of the person. In their systematic review, Ma et al. [47] identify an overall pooled prevalence of CRF of 52%. CRF varies with time, and is more significant close to the end of treatment and can affect cancer survivors up to 15 years after diagnosis [48].

The variety and ever-changing difficulties affecting cancer survivors emphasize the need for cancer survivors to have access to personalized, holistic, and integrated follow-up care [49]. Cancer navigation could be an interesting way of meeting all these goals and optimizing care and support throughout the cancer continuum [50]. Knowing the needs of cancer survivors appears to be the starting point for such follow-up care. This underscores the importance of conducting systematic assessments of supportive care needs, as part of the care routine [14,20,21,23,26,28,51] as soon as acute treatment ends. The World Health Organization [52] and the National Comprehensive Cancer Network [53] support these kinds of assessments as an essential step for managing the late effects of cancer and/or treatments. Recently, more guidelines have been developed that recommend assessments, but there are not enough yet, according to Hahn et al. [44]. Health professionals also have a role to play in sustaining survivors in their follow-up journey. According to William et al. [31], “effective communication could involve women being encouraged to ask questions, feeling listened to and not feeling rushed” and “poor communication left them feeling trivialized and uncared for”. This is emphasized by the study from Chamber et al. [35], where advanced prostate cancer survivors described their inability to raise their concerns and to receive clear answers to questions due to insufficient and selective communication from clinicians. Consequently, they must find solutions by themselves.

Female gender, young age, lack of family or social support and a high level of education are considered as risk factors, while employment status, relationship status or geographical location can have a positive or negative influence on needs. For all these socio-demographic factors, the level of evidence varies between the authors, but these findings are confirmed by other studies, notably the qualitative systematic review of Bellas et al. [51]. These authors also consider other factors like type of cancer, treatments received, culture, language, and the presence of comorbidities.

Although the concept of moving into survivorship is not clearly defined [7], it is certain that this transition phase is critical for people and their families [11]. At this point, they receive less support than during the treatment period [6], while suffering the side-effects of the treatment or illness. They have a higher prevalence of unmet needs [22], and they often feel abandoned [19,25] and perceive a lack of access to care. This would, therefore, seem to be a good starting point for a systematic needs’ assessment, although it is also advisable to do this beforehand [52]. This assessment could allow to provide an individual, holistic, and integrated follow-up. This approach will enable the patient and the professional to identify what is really important and what needs to be addressed to help the person affected making this transition to survivorship, boost their self-esteem, enable them to reintegrate more effectively into society and the workplace, avoid precariousness and, ultimately, improve their quality of life. A survivorship care plan should be developed from the data collected from this needs’ assessment to provide “practical guidance” [11].

Lastly, it is important to point out that the experience of cancer can also lead to positive changes [25]. As described by Tedeschi and Calhoun [54], “post-traumatic growth refers to positive psychological change experienced as a result of the struggle with highly challenging life circumstances”. Many cancer survivors, from 50% to more than 80%, report some benefits such as “strengthened interpersonal relationships, commitment to life priorities, life appreciation, personal regard, spirituality, and attention to health behaviours” [55]. One hypothesis for these changes is that the proximity of illness and death is an opportunity to take stock and redefine one’s priorities in life, leading to changes that are seen as positive. However, these benefits appear to diminish over time [55].

### 4.2. Review Strengths and Limitations

The principles of the approach (recommendations) developed by the Joanna Briggs Institute [12] for scoping studies were followed for this review, minimizing methodological failures. Only peer-reviewed systematic reviews were included, which optimizes the quality of findings. Using the AMSTAR checklist for quality appraisal [18] ensured that we would avoid the worst-quality studies. This review focuses on all types of cancer in developed countries. Although the contexts are different for each study, these findings can be generalized, which is useful for healthcare professionals to understand the overall needs of survivors and offer them appropriate care.

However, some limitations of our review must be acknowledged. This review is centered on high-income countries (HIC) and, as such, the findings may not be directly applicable to low- and middle-income countries (LMIC). Only four databases were used for the research. Moreover, grey literature was not searched. Some articles and information may, therefore, be missing. Although we cannot rule out the possibility that studies have been published since January 2023, the risk that new knowledge may have emerged between the date of the last electronic search and the publication of this article is low. Finally, due to the heterogeneity of the selected studies, it was impossible to assess the prevalence of needs. This would have helped the prioritization of needs.

## 5. Conclusions

People with a history of cancer often experience various side effects of the illness or treatments, which can also lead to positive outcomes. This review identifies a wide range of needs in the long-term follow-up of adult cancer survivors, and provides a new classification. Although the evidence is not conclusive, there are some indications that these needs are particularly pronounced directly after the completion of the initial treatments, and that the needs vary depending on the socio-demographic factors. Cancer survivors, their relatives, and health professionals face a multifactorial problem, and often have difficulties to identify and prioritize needs, which are highly individual and evolving. Therefore, needs should be systematically and regularly assessed.

## Figures and Tables

**Figure 1 healthcare-12-01058-f001:**
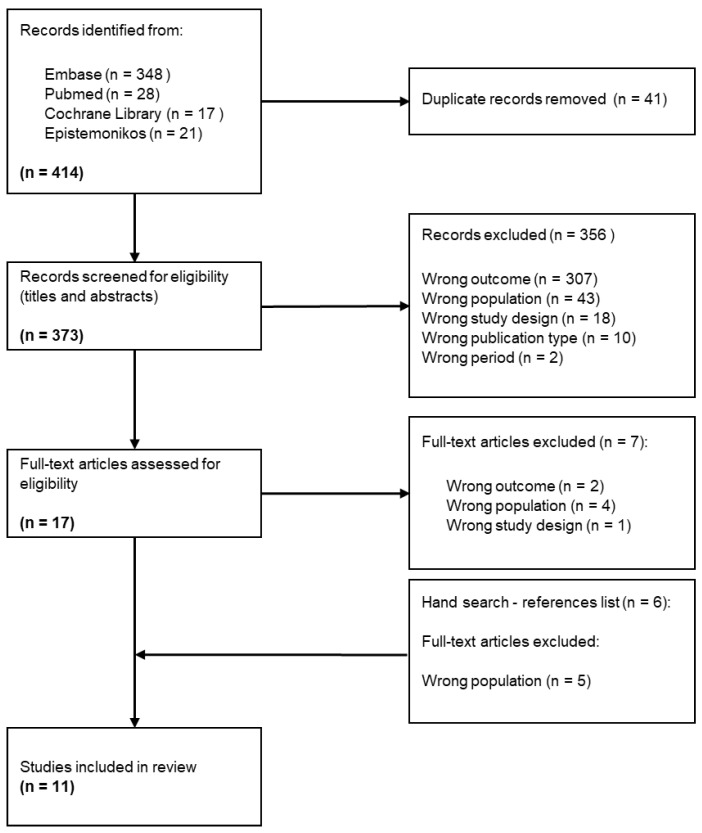
PRISMA flow chart of the article selection process.

**Table 1 healthcare-12-01058-t001:** Characteristics of the included studies: critical appraisal within sources of evidence.

Studies	Characteristics of the Studies			Context	Population			Limitations	Critical Appraisal
	Aim of the Studies	Number of Included Articles	Period of Analysis (of the Studies)	Countries	Number of Participants (Range)	Gender	Cancer Type		
Dahl et al. (2013) [23]	To investigate knowledge on the quality of life after cancer, which factors could be predictors, and knowledge on gynecological cancer patients’ needs and preferences regarding follow-up.	57	1995–2012	Not reported	Not reported	Not reported	Gynecological	▪ No limitations are reported; ▪ Little or no information about the target group and setting; ▪ Prisma flow chart is not available; ▪ Only capture English research.	Not sufficient quality
Hoekstra et al. (2014) [24]	To report how adult cancer survivors describe their care needs in the general practice environment.	15	1990–2012	▪ UK; ▪ USA; ▪ Canada; ▪ Denmark; ▪ Italy.	970 (6–431)	Men Women	▪ Bladder; ▪ Prostate; ▪ Breast; ▪ Colorectal; ▪ Head and neck; ▪ Lung; ▪ Melanoma; ▪ Testis; ▪ Gynecological; ▪ Bowel; ▪ Hematological; ▪ Non-Hodgkin’s lymphoma; ▪ Hodgkin’s; ▪ Gastrointestinal; ▪ Unknown/other.	▪ Use of only 3 databases for the search; ▪ Delay between the search and the publication; ▪ Possibly under representation of all existing needs.	Good quality
Hyun et al. (2016) [21]	To examine the unmet information needs and the unmet psychosocial support needs of adult thyroid cancer survivors.	7	2008–2016	▪ USA; ▪ Canada; ▪ Netherlands; ▪ South Korea.	6215	Majority of women	Thyroid	▪ Level of agreement between reviewers was limited; ▪ No stratification of needs according to important variables (clinic-histopathologic sub-group, life stage, or disease status in response to treatment); ▪ Only capture English research.	Sufficient quality
Kotronoulas et al. (2017) [19]	To synthesize evidence in relation to the supportive care needs of people living with and beyond cancer of the colon and/or rectum.	45	1996–2016	▪ UK; ▪ Australia; ▪ Other (not specified).	10,057 (5–3011)	Men (64.5%) Women (35.5%)	Colon and/or rectum	▪ Mixed patient samples; ▪ No grey literature researches; ▪ Limitations due to the tool used for appraising the methodological quality; ▪ Only capture English research.	Good quality
Lehmann et al. (2021) [14]	To identify the prevalence of sexual health-related care needs and the types of needs that should be addressed by providers.	35	2004–2019	▪ Denmark; ▪ USA; ▪ Germany; ▪ Canada; ▪ Australia; ▪ Netherlands.	5938 (8–879)	Majority of women	▪ Breast; ▪ Testicular; ▪ Gynecological.	▪ Focus on need addressed by professionals. ▪ Risk of biased assessment of all the included studies. ▪ Only capture English research.	Sufficient quality
Lim et al. (2021) [25]	To synthesize the current body of qualitative research on colorectal cancer survivorship as early as the immediate post-operative period, and to compare the experiences of early stage and advanced colorectal cancer survivors.	81	2006–2019	▪ USA; ▪ Europe; ▪ UK; ▪ Australia; ▪ Asia; ▪ Canada; ▪ New Zealand; ▪ Middle East.	Not reported	Not reported	Colon and/or rectum	▪ Search was not exhaustive; ▪ Subjective inclusion of articles due to differing definition of “survivorship” and lacked clarity on participants “survivorship status”; ▪ Deviation from the original PROSPERO protocol; ▪ Only capture English research.	Sufficient quality
Lisy et al. (2019) [26]	To identify the most prevalent unmet needs of cancer survivors in Australia and to identify demographic, disease, or treatment-related predictors of unmet needs.	17	2007–2018	Australia	Not reported	Not reported	▪ Gynecological; ▪ Breast; ▪ Brain; ▪ Hematological; ▪ Endometrial; ▪ Prostate; ▪ Testicular; ▪ Various.	▪ Data are limited by the measure used to assess unmet needs; ▪ Review does not include a proportionate distribution of cancer types in Australia; ▪ Study selection and quality appraisal were conducted primarily by one reviewer; ▪ Studies included in this narrative review were equally weighted regardless of sample size.	Sufficient quality
Maguire et al. (2015) [20]	To synthesize evidence with regard to the supportive care needs of women living with and beyond cervical cancer.	14	1990–2013	▪ USA; ▪ Canada; ▪ UK; ▪ Indonesia; ▪ South Korea; ▪ Nigeria; ▪ Thailand.	1414 (10–968)	Women	Cervical	▪ Search was limited to the most common databases; ▪ No grey literature research; ▪ Only capture English research.	Good quality
Miroševič et al. (2019) [22]	To determine the prevalence and identify the factors that contribute to higher levels of the unmet needs. To identify the most commonly unmet needs and those factors that contribute to higher levels of unmet needs in each domain separately.	26	2007–2015	▪ Australia; ▪ UK; ▪ USA; ▪ China; ▪ Singapore; ▪ Canada; ▪ Ireland; ▪ Netherlands; ▪ Iran; ▪ South Korea.	10,533 (63–1668)	Men Women	▪ Breast; ▪ Gynecological; ▪ Hematological; ▪ Head and neck; ▪ Colorectal; ▪ Endometrial; ▪ Various.	▪ Most of the included studies were cross-sectional; ▪ Included studies lacked information (prevalence, factors associated with specific domains, stage of cancer at diagnosis); ▪ Homogenous sample in several studies; ▪ Only capture English research.	Sufficient quality
Pape et al. (2021) [27]	To describe the experiences and needs of patients with rectal cancer confronted with bowel problems after stoma reversal.	10	2006–2021	▪ UK; ▪ USA; ▪ China; ▪ Taiwan; ▪ Sweden; ▪ Netherlands.	156 (5–36)	Men (approx. 58%) Women (approx. 42%)	Rectal with Stoma reversal	▪ Some studies do not reach data saturation; ▪ Small sample for some studies; ▪ Some studies were performed as single center studies; ▪ Most of the studies did not report on the severity of participants’ bowel problems.	Good quality
Van der Kruk et al. (2021) [28]	To review levels of psychosocial morbidity and the experiences and needs of people with cancer and their informal caregivers, living in rural or regional areas.	65	2010–2021	▪ Australia; ▪ USA; ▪ Canada; ▪ Europe.	Not reported	Not reported	▪ Breast; ▪ Hematological; ▪ Colorectal; ▪ Lung; ▪ Head & neck; ▪ Gynecological; ▪ Prostate; ▪ Myeloma; ▪ Various.	▪ Included studies have different definition of “rurality”; ▪ Different methodological approach and data sources of the studies; ▪ No meta-analysis was conducted due to the heterogeneity of the studies; ▪ Findings are conceptual rather than statistical; ▪ Only capture English research.	Sufficient quality

**Table 2 healthcare-12-01058-t002:** Details of domains and categories of needs.

Domains of Needs	Definition	Categories of Needs	Examples of Challenges	Number of Studies Reporting These Needs
Health-related information	Need to receive and process adequate information on all types of subjects to meet certain objectives.	Access	Lack of information (all kinds of information), quality and delivery of information.	10
		Education	Difficulties to process information, comprehension, and quality assessment.	
Health system	Need to access a personalized, comprehensive, and integrated care and support pathway to reduce or treat the consequences of the disease and/or treatments.	Healthcare professionals	Lack of knowledge of the unique needs of rural survivors by medical staff located in metropolitan treatment centers, on-going patient–clinician contact, post-operative follow-up (hospital doctor) or post-treatment follow-up (specialist nurse), helping with common (late) treatment effect, initialization of discussions about sexual health by providers, empathic and sensitive discussion on sexual health, overcome taboos, enough time to discuss sensitive matters.	10
		Health and supportive care	Coordination of health care services (primary and secondary), access to counselling/support groups, access to complementary/alternative medicine, gap in supportive care, medical help/treatment for non-cancer related problems, general preventative healthcare, access and continuity of care, comprehensive care, regular monitoring of needs, navigation in health system.	
Mental	Supportive care needs to reduce emotional, existential, interpersonal and/or psychic health conditions, due to illness and/or treatment, that disrupt a person’s behavior or reasoning.	Emotional	Deal with altered body image, appearance (attractiveness, self-image, desirability, femininity), emotional health.	9
		Existential	Fear of cancer recurrence, uncertainty, adversity, lifestyle changes, worries about the future.	
		Interpersonal/intimacy	Changes in sexuality, coping with sexual dysfunction, lack of sexual desire, anxiety about sexual intercourse, feeling to be forced to fulfill the partner’s sexual desires (cultural pressure and expectation).	
		Psychic	Stress, feeling of abandonment after treatment, anxiety, distress, depression.	
Practical	Need for support to limit the impact of the disease and/or treatment on daily life.	Daily activities	Not being able to do usual things, transportation, identification, and integration of health behaviors.	7
		Financial impact	Financial well-being, worry about earning money, fighting financial toxicity.	
		Work	Return to work, adapting work to new capacities (position, schedule, workload, etc.), change of professional activity, reactions of colleagues/leaders.	
Relationship	Need for support to reduce or deal with the consequences of the illness and/or treatments that disrupt interactions with the family and the social environment.	Family	Support of family for its own worries, family’s future, worry about partners and family.	6
		Social	Embarrassment in social situations, relationships with others, lack of practical and emotional support from peers and the environment, difficulties and tensions in relationships, isolation, social role change, social desirability.	
Physical	Supportive care needs to alleviate or treat the physical and cognitive consequences of the disease and/or treatments.	Body	Fatigue/lack of energy, pain, physical problems, dysfunction, sleep loss, urinary incontinence, bowel dysfunction, difficulty breathing, infertility, hormone changes, loss of strength, nausea/vomiting, neuropathy, sexual dysfunction, skin irritation, weight changes, infected or bleeding wound, mouth- or eye-related, physical examination, managing side-effects (physical symptoms).	6
		Cognitive	Memory loss, difficulty concentrating or cognitive dysfunction.	

## Data Availability

The authors confirm that all data analyzed during this study are included in this published article.

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
