# Peer review of "Domains and Categories of Needs in Long-Term Follow-Up of Adult Cancer Survivors: A Scoping Review of Systematic Reviews"

_healthcare, 2024, doi:10.3390/healthcare12111058_

Round 1

Reviewer 1 Report

Comments and Suggestions for Authors

very interesting this idea of yours

i was wondering if a timing for these needs could be identified based on your literature search (ie for example physical need more prominent after the first 3 months, social needs during the first month etc). also any relationship between the concomitancy of various need groups and the type of cancer? or subsequent survival?

Author Response

Dear Reviewer 1,

Thank you for taking the time to review our scoping review manuscript titled "Domains and Categories of Needs in Long-term Follow-up of Adult Cancer Survivors: A Scoping Review of Systematic Reviews.". We appreciate your insightful feedback, which has greatly contributed to the enhancement of our work.

Attached, you will find the revised version of the manuscript, where we have addressed each of your comments and suggestions in detail. Below, we provide a response to your question:

  1. Your question is indeed interesting and pertinent. We revisited the literature to delve deeper into this aspect. While some data exist that underscore the significance of needs during the transitional phase (as referenced in the article, refer to lines 458-462), other sources have identified specific "periods" for certain needs. However, due to the highly individual nature of the survivorship journey, it proves challenging to generalize this data and pinpoint exact timings. The same complexity applies to the correlation between need groups and the type of cancer survived.
  2. We are currently analyzing data from the Swiss Cancer Patient Experiences study, and we will certainly consider these questions during our analysis.

Should you have any further questions or require additional clarification, please do not hesitate to contact us.

Thank you once again for your valuable feedback and constructive suggestions. 

Best regards,

Nicolas Sperisen

Reviewer 2 Report

Comments and Suggestions for Authors

The reviewed manuscript present a solid piece of work on a very pertinent and difficult area of research. The methods used are adeqaute, Limitations are discussed. I would recommend the authors to provide the results of their critical apprasial of included systematic reviews together with their characteristics, which would make the table more informative (Table 1). It would also help a reader a lot if tables would be reorganised to be neat, numbered, without extra spaces that take time and resource to get to the essense of the content. Conclusions present more the continuation of discussion, which is already elaborate enough. In my view , the conclusion and abstaract need more work to make them structured (I prefer bulleted), concise and catchy.  

Author Response

Dear Reviewer 2,

Thank you for taking the time to review our scoping review manuscript titled "Domains and Categories of Needs in Long-term Follow-up of Adult Cancer Survivors: A Scoping Review of Systematic Reviews.". We appreciate your insightful feedback, which has greatly contributed to the enhancement of our work.

Attached, you will find the revised version of the manuscript, where we have addressed each of your comments and suggestions in detail. Below, we provide a summary of the changes made:

  1. We have completed Table 1 (refer to lines 208, 211) with the results of the critical appraisal. Additionally, we have made edits to improve the readability of the tables, and we have relocated Table 2 to appendix C (refer to lines 220, 531). The final layout will be handled by the journal.
  2. The structure of the abstract is determined by the journal's guidelines, leaving us little room for adjustments. However, we have simplified the abstract by removing certain sections (see lines 18, 27-28) to enhance clarity.
  3. Regarding the conclusion, we have transferred the section discussing follow-up and systematic need assessment to the discussion section (refer to lines 464-469, 510-517). As a result, the conclusion is now more concise. We have opted not to utilize bullet points in this case, as we do not believe they would enhance readability.

We believe that these revisions have significantly improved the clarity, coherence, and overall quality of the manuscript. We are grateful for the opportunity to address your concerns and refine our work accordingly.

Should you have any further questions or require additional clarification, please do not hesitate to contact us.

Thank you once again for your valuable feedback and constructive suggestions. 

Best regards,

Nicolas Sperisen

Reviewer 3 Report

Comments and Suggestions for Authors

Abstract: In general, the abstract is well written. I would omit the abbreviation JBI. Authors should state what kind of study they performed. The part about the Swiss health care system should be omitted. I would recommend “high-income countries”
Key words are appropriate
Introduction: Reference 4 should be omitted about Switzerland. Paragraph 2 should be expanded because it is a basis of the study taken.
Objectives: Cancer survivorship is a global problem, not just in Switzerland. The authors need to emphasize this part and avoid the particularities. Instead, they should rename the objective into high-income countries. The research questions are well defined
Material and methods
Eligibility criteria are appropriate as well as inclusion and exclusion criteria as well the search strategy.
Results: PRISMA flow chart and table 1. do not need revision. Table 3. (domains, need) is a bit confusing and should in short be explained in the text of the manuscript. When reporting domains and included study the authors should critically analyze the strengths and weaknesses of the studies included.
Discussion is clear, has a rationale and no revision is needed as well the limitations
Over all minor revisions are needed prior to the acceptance of the article. 

Author Response

Dear Reviewer 3,

Thank you for taking the time to review our scoping review manuscript titled "Domains and Categories of Needs in Long-term Follow-up of Adult Cancer Survivors: A Scoping Review of Systematic Reviews". We appreciate your insightful feedback, which has greatly contributed to the enhancement of our work.

Attached, you will find the revised version of the manuscript, where we have addressed each of your comments and suggestions in detail. Below, we provide a summary of the changes made in response to your feedback:

Abstract:

  • We have removed the abbreviation JBI (refer to line 18)
  • We have removed the section discussing the Swiss healthcare system (refer to lines 27-28). Instead of replacing it with "high-income countries," we have opted to proceed without specifying a broader category, as we cannot generalize that healthcare systems in other countries are not designed to support cancer survivors.

Introduction:

  • We have removed reference 4 (refer to line 45).
  • We have expanded the second paragraph by providing more detailed information (refer to lines 49-50, 51-53, 59-63).

Objectives:

  • We have removed the section discussing Switzerland and our aim to develop a screening tool (refer to lines 69-73).

Results:

  • We have included a description for Table 3 (Table 2 in the current version, refer to lines 224-226).
  • Table 1 includes information about the limitations of the studies included. Moreover, in response to feedback from another reviewer, we have incorporated our critical appraisal into Table 1 (refer to line 208). We are confident that this adequately represents the strengths and weaknesses of the studies and does not require supplementation.

We believe that these revisions have significantly improved the clarity, coherence, and overall quality of the manuscript. We are grateful for the opportunity to address your concerns and refine our work accordingly.

Should you have any further questions or require additional clarification, please do not hesitate to contact us.

Thank you once again for your valuable feedback and constructive suggestions. 

Best regards,

Nicolas Sperisen

Reviewer 4 Report

Comments and Suggestions for Authors

The scoping review of systematic reviews titled “Domains and categories of needs in long-term follow-up of adult cancer survivors: A scoping review of systematic reviews” by Sperisen, N., et al., is a thorough analysis conducted according to the recommendations of the Joanna Briggs Institute.   The authors have summarized the literature (11 eligible papers from a total of 414 articles) pertaining to the needs and the factors that influences these needs.  Authors have categorized the needs into six domains, viz., (a) health-related information; (b) health system; (c) mental; (d) practical; (e) relationship and (f) physical and 15 categories. Analysis of the collected information showed that lack of adequate information and the lack of access and/or continuity of supportive care were the most prominent needs. Key risk factors identified in this scoping review are (a) Female gender; (b) Younger age; (c) low level of family and/or social support, and (d) higher educational level.  Authors have mentioned that the current Swiss healthcare system is not designed to support the cancer survivors.  Authors have concluded that the assessment of needs should be systematically conducted, recorded and provided for a better awareness on the part of health professionals and to allow an individual, holistic, and integrated follow-up.

Although it is an interesting study it focuses on “high-income” countries (context), hence the findings are applicable only such countries.  But, cancer incidence and mortality rates are much higher in low- and middle-income countries (LMICS).  It is one of the limitations of this study.

Suggestions

The domains and categories of needs can be represented as a figure

Authors can also think of condensing the tables, as they are very large and exhaustive

Author Response

Dear Reviewer 4,

Thank you for taking the time to review our scoping review manuscript titled "Domains and Categories of Needs in Long-term Follow-up of Adult Cancer Survivors: A Scoping Review of Systematic Reviews". We appreciate your insightful feedback, which has greatly contributed to the enhancement of our work.

Attached, you will find the revised version of the manuscript, where we have addressed each of your comments and suggestions in detail. Below, we provide a summary of the changes made in response to your feedback:

  1. We completely agree with your perspective and have added the non-representativeness of our results for low- and middle-income countries as a limitation (refer to lines 493-495).
  2. The suggestion of having a figure instead of Table 3 (Table 2 in the current version) for representing needs is appealing. However, we believe that important information, such as the definition of domains, examples of needs, and the number of studies reporting these needs, would be omitted. Therefore, we prefer to retain the table format. An alternative option could be to include a graphical abstract (as in the preprints publication), which could summarize the needs. 
  3. We concur with the observation regarding the complexity of the tables. Despite this, we have opted to complete Table 1 in response to another reviewer's request. However, to manage overload, we have edited all the tables and relocated Table 2 to Appendix C (refer to lines 220, 531). The final layout will be handled by the journal.

We believe that these revisions have significantly improved the clarity, coherence, and overall quality of the manuscript. We are grateful for the opportunity to address your concerns and refine our work accordingly.

Should you have any further questions or require additional clarification, please do not hesitate to contact us.

Thank you once again for your valuable feedback and constructive suggestions. 

Best regards,

Nicolas Sperisen